# Effect of Minimally Invasive Spine Stabilization in Metastatic Spinal Tumors

**DOI:** 10.3390/medicina58030358

**Published:** 2022-03-01

**Authors:** Kazuo Nakanishi, Kazuya Uchino, Seiya Watanabe, Kosuke Misaki, Hideaki Iba

**Affiliations:** Department of Orthopedics, Traumatology and Spine Surgery, Kawasaki Medical School, Okayama 701-0192, Japan; kazuya_u@med.kawasaki-m.ac.jp (K.U.); seiya-w@med.kawasaki-m.ac.jp (S.W.); k-misaki@med.kawasaki-m.ac.jp (K.M.); i-hideaki@med.kawasaki-m.ac.jp (H.I.)

**Keywords:** metastatic spinal tumor, minimally invasive spine stabilization, percutaneous pedicle screw, skeletal-related event

## Abstract

*Background and Objectives*: There have been numerous advances in spine surgery for metastatic spinal tumors, and minimally invasive spine stabilization (MISt) is becoming increasingly popular in Japan. MISt is a minimally invasive fixation procedure that temporarily stabilizes the spine, thereby reducing pain, preventing pathological fractures, and improving activities of daily living at an early stage. MISt may be useful given the recent shift toward outpatient cancer treatment. *Materials and Methods*: This study enrolled 51 patients with metastatic spinal tumors who underwent surgery using MISt between December 2013 and October 2020. The Spinal Instability Neoplastic Score, an assessment of spinal instability, was used to determine the indication for surgery, and the Epidural Spinal Cord Compression scale was used for additional decompression. *Results*: The patients comprised 34 men and 17 women, and the mean age at surgery was 68.9 years. The mean postoperative follow-up period was 20.8 months, and 35 of 51 patients (67%) had died by the last survey. The mean operative time was 159.8 min, mean blood loss was 115.7 mL, and mean time to ambulation was 3.2 days. No perioperative complications were observed, although two patients required refixation surgery. Preoperatively, 37 patients (72.5%) were classified as Frankel grade E. There were no cases of postoperative exacerbation, and six patients showed improvement of one or more Frankel grades after surgery. The median duration of patient survival was about 22.0 months. Patients with breast, prostate, renal, and thyroid cancers had a good prognosis, whereas those with gastrointestinal and head and neck cancers had a poor prognosis. *Conclusions*: MISt can benefit patients who are ineligible for conventional, highly invasive surgery and is also suitable because cancer treatment is increasingly performed on an outpatient basis. Furthermore, choosing the right surgery for the right patient at the right time can significantly affect life expectancy.

## 1. Introduction

Cancer treatment has changed drastically over the past decade. Advances in therapies include molecular-targeted drugs, immune checkpoint inhibitors, bone-modifying agents, chemotherapy, and radiotherapy. Multidisciplinary and systemic treatments have also started being used. These developments have significantly improved life expectancy, and we are now living in an era of coexistence with cancer, with shorter hospitalization periods and a shift of treatment to outpatient settings.

The effects of metastatic spinal tumors differ from those of other bone metastases. The occurrence of skeletal-related events (SREs) [1,2], such as fractures and paralysis, can significantly impair activities of daily living (ADLs) and reduce quality of life (QOL). If ADLs are impaired, patients may not be eligible for chemotherapy or radiotherapy, which may affect their life expectancy [3,4] and hinder the transition of treatment to an outpatient setting.

Conventional spinal surgery for metastatic spinal tumors is highly invasive and requires a long hospital stay to stabilize the spine and relieve nerve compression caused by the tumor. In recent years, however, there have been significant advances in spinal surgeries. Minimally invasive surgeries and locally curative surgeries such as total en bloc spondylectomy (TES) [5] have become more common. Minimally invasive spine stabilization (MISt) using percutaneous pedicle screw (PPS) fixation of the spine, which has been performed relatively frequently in Japan since 2005, is indicated for cancer-bearing patients with impaired systemic reserves. In this context, MISt is a minimally invasive fixation technique that uses PPSs to temporarily stabilize the spine, thus reducing pain and facilitating early improvement in ADLs. It can also prevent pathological fractures and is useful given the current transition to outpatient treatment. This article describes the efficacy of the MISt procedure in patients with metastatic spinal tumors.

## 2. Materials and Methods

This study was a retrospective study, conducted according to the guidelines of the Declaration of Helsinki and approved by the Institutional Review Board of our hospital (ID: 5054-01, 20 October 2020). A total of 51 consecutive patients with metastatic spinal tumors underwent surgery at our hospital from December 2013 to October 2020.

### 2.1. Indication for Surgery

We provided multidisciplinary therapeutic intervention for all metastatic spine tumors [6]. Once metastasis was detected in the spine, the spinal surgeon intervened to determine the treatment plan. Eligibility for surgery was determined by comprehensive patient evaluations performed by attending doctors and specialists and was also based on prognosis. In general, for patients without paralysis whose main symptom was pain, radiotherapy was the first-line treatment. In order to clarify the indication for surgery, spinal instability was evaluated using a spinal instability neoplastic score (SINS) [7] at our hospital. Surgery was recommended for patients with SINS score ≥13. In SINS, the indication for surgery in cases of imminent instability of 7 to 12 points became a problem. For impending spinal instability defined by a SINS score between 7 and 12, surgery was selected if imaging studies showed vertebral collapse or osteolysis of the posterior wall of the vertebral body or the pedicle of the vertebral arch or if ambulation was impossible because of severe pain. In principle, surgery was indicated for patients with a well-controlled primary tumor or for whom there were additional treatment options.

### 2.2. Surgical Technique

Preoperative evaluation and careful preoperative planning were considered important. Cancer is a systemic disease, and the first step was to evaluate the general condition of the patient. Surgery was contraindicated for bone marrow carcinosis, a condition that causes abnormalities of the blood and hematopoietic organs due to metastasis of tumor cells to the bone marrow. MRI was used to evaluate tumor spread to the vertebral bodies, and bone scintigraphy was used to evaluate systemic metastasis. Since patients with multiple spinal metastases are inevitably targeted, long fusion was often used. The range of spinal fusion and the size of the screw were decided in advance, so as to reduce the durations of surgery and radiation exposure.

Surgery was conducted according to the standard MISt procedure with commonly used PPSs [8]. The fusion range was basically three vertebrae above and two below. In principle, screws were not inserted into metastatically involved vertebral bodies. Additional decompression was performed in patients with neurological symptoms and those with grade ≥2 lesions based on the Epidural Spinal Cord Compression scale [9]. During decompression, the implant was placed first, and then all wounds were closed. The decompression procedure was then performed with a separate skin incision in the midline. Bone grafting was generally not performed.

The outcome and the following items were evaluated: (1) operative time; (2) blood loss; (3) time to ambulation; (4) Tokuhashi score [10] and New Katagiri score [11] to evaluate prognosis; (5) SINS to evaluate the stability of the vertebral body [7]; (6) Frankel grade to assess the degree of spinal cord injury; and (7) Eastern Cooperative Oncology Group Performance status (PS) [12] to evaluate QOL. The association of each parameter with the outcome was assessed using the Mann–Whitney U test, and survival time was calculated from the date of enrollment to the date of death using the Kaplan–Meier method. Statistical analyses were conducted using SPSS software, version 23.0 (IBM Japan Business Services Co., Ltd., Tokyo, Japan), with a significance level of 5%.

## 3. Results

Table 1 shows the patient characteristics. The mean age of all subjects (n = 51; male, n = 34; female, n = 17) was 68.9 ± 9.9 years (range: 44–81 years). The primary tumors were breast cancer (n = 11), lung cancer (n = 10), prostate cancer (n = 7), gastrointestinal cancer (n = 5), renal pelvis/ureter cancer (n = 4), kidney cancer (n =3), thyroid cancer (n = 2), uterine cancer (n =2), multiple myeloma (n = 2), primary unknown cancer (n = 2), pancreatic cancer (n = 1), head and neck cancer (n = 1), and others (n =1). The mean postoperative follow-up period was 20.8 ± 21.8 months (range: 1–87 months), and 35 of 51 (67%) patients died before the final follow-up. Follow-up was successfully completed in 44 of 51 (86.3%) patients.

With regard to prognostic prediction scores, the mean Tokuhashi score was 9.3 ± 2.6 (range: 5–14), and the mean New Katagiri score was 4.6 ± 2.0 (range: 1–9).

For each patient, the treatment strategy was determined based on the total score of the six SINS items, including lesion location, pain, and vertebral collapse [7]. Given a maximum score of 18 points, a score of ≤6 points was classified as stable condition, 7 to 12 points was impending spinal instability, and ≥13 points was instability. The mean SINS score was 9.4 ± 2.8 (range: 3–16).

Table 2 displays the surgical details. The mean number of fixed intervertebral segments was 6 (range: 2–16). Additional decompression was performed in 14 patients. The mean operative time was 159.8 ± 65.4 min (range: 52–410 min), mean blood loss volume was 115.7 ± 126.7 mL (range: 10–630 mL), and mean time to ambulation (excluding three patients) was 3.2 ± 2.2 days (range: 1–14 days). The relationship between additional decompression and the SINS score is shown in Table 3. Two-thirds of unstable patients with a score of ≥13 points underwent additional laminectomy. Most patients with a score <12 points underwent temporary fixation only.

No systemic complications were observed in the perioperative period, although three patients required further surgery. Two patients had loosened screws, and one patient had metastases detected in other parts of the body, each of whom underwent additional surgery. As concomitant treatments, 46 (90%) patients received radiotherapy (RT) and 41 (80%) received a bone-modifying agent.

Table 4 shows the preoperative and postoperative Frankel grades. 37 (72.5%) patients had a preoperative Frankel grade of E (no paralysis). Three patients were classified as Frankel grade A, four as grade C, and seven as grade D. There were no cases of postoperative exacerbation, and six patients improved postoperatively by at least one grade.

The mean PS was 2.5 ± 1.3 before surgery, which significantly improved to 1.7 ± 1.3 after surgery (*p* < 0.01). 31 patients were discharged home (mean time from surgery to discharge: 31.9 ± 18.0 days), and 17 patients were transferred to another hospital (mean time from surgery to transfer: 39.4 ± 7.0 days).

Among the 51 patients receiving MISt, the median duration of survival was about 22.0 ± 8.6 months (95% confidence interval [CI] 5.1–38.8; Figure 1).

We analyzed the characteristics of the eight patients who experienced early death, defined as occurring within 3 months postoperatively (Table 5). Four of these patients were first seen after the onset of SRE. The type of carcinoma varied, and the mean SINS score was 10.1 ± 3.8 points (range: 3–16), indicating imminent instability. The mean preoperative PS was 2.4 ± 1.2 (range: 0–4), and the mean postoperative PS was 2.0 ± 1.6 (range: 0–4; *p* > 0.05). Three patients were discharged home, two were transferred to another institution, and the others died during hospitalization. Surgery was not the direct cause of death in any of the cases.

Table 6 shows the mortality by cancer type. The prognosis was favorable in patients with breast, prostate, kidney, or thyroid cancer but was poor in those with cancers of the digestive system or head and neck. Therefore, the underlying disease should be considered when determining the eligibility of patients for surgery.

### Case Presentation

A 63-year-old woman with metastatic spinal tumors secondary to breast cancer (Figure 2) was referred for weakness of the left lower extremity (Frankel grade D) and gait disturbance. Multiple metastases in the ribs and pelvis were also observed. The Tokuhashi score was 10 points, the New Katagiri score was 6 points, and the PS was 4. Decompression surgery was performed at T7–8, and screws were inserted and fixed in the unaffected vertebrae (C3–L2). The operation time was 410 min, and blood loss was 210 mL The patient began walking on the second postoperative day, and radiotherapy was started on postoperative day 15. The patient’s paralysis of the lower limbs improved, PS improved to 2, and she was discharged home on postoperative day 60. The patient died 3 years 8 months after the surgery, but no loosening of the screw was observed until the patient’s death.

## 4. Discussion

Due to advances in cancer therapies, the prognosis of patients with cancer has improved. Thus, there is a need to extend the healthy life expectancy of patients with metastatic spinal tumors by maintaining ADLs and enhancing QOL. Previously, conventional surgery for metastatic spinal tumors has consisted of spinal decompression and fixation, which required a large incision. As a result, this surgery necessitated a long hospitalization and postoperative bed rest and delayed the transition to other treatments such as radiation therapy and chemotherapy. As conventional surgery is highly invasive, patients’ eligibility for surgery depends on their systemic reserves and life expectancy. Ineligible patients often develop SREs [1,2], such as pathological fracture and paralysis. Significant impairment of ADLs and QOL due to SREs leads to ineligibility for radiotherapy and chemotherapy and consequently affects life expectancy.

For patients with spinal metastases who have good PS, who have no metastases in the principal organs, and in whom the tumors are localized to the vertebral bodies, curative surgery such as TES [5] is preferable. However, only a small number of patients are eligible for this procedure. In many cases, the only option is to perform minimally invasive palliative surgery [10]. Although no eligibility criteria for life expectancy have been established with regard to surgery for metastatic spinal tumors, some studies have suggested that conventional, highly invasive surgery requires a tolerable general condition and a life expectancy of 6 months or longer [5,10].

MISt is performed for treating trauma, infection, metastatic spinal tumors, and scoliosis. In particular, it is the best treatment option for patients with metastatic spinal tumor who have a limited life expectancy [8,13,14,15]. As MISt allows for early postoperative ambulation and discharge, its indication has been extended to patients with a life expectancy of less than 6 months. Furthermore, MISt allows for early postoperative adjuvant therapies, such as radiotherapy and chemotherapy.

Based on our experience with MISt to date, the technique demonstrates three advantages for the treatment of metastatic spinal tumor:MISt is less invasive than conventional surgery. This leads to reduced bleeding and not only allows patients to recover from surgery earlier but also makes it possible to perform surgery in patients who are ineligible for conventional surgery because of poor systemic reserves. Miscusi et al. [16] found that the mean operative times was 3.2 h in patients treated with the conventional method versus 2.2 h in those undergoing minimally invasive spine surgery (MISS; *p* < 0.01); the mean intraoperative blood losses were 900 mL and 240 mL, respectively, and the mean durations of postoperative bed rest were 4 days and 2 days (*p* < 0.01), respectively. No patients in the MISS group required blood transfusions. Watanabe et al. [17] studied the results of 112 patients who underwent conventional surgery and who differed in terms of their preoperative ambulatory status. The ambulatory and non-ambulatory groups had mean intraoperative blood losses of 960 mL (20–3420 mL) and 1230 mL (230–3090 mL), respectively, and mean operative times of 199 min (60–975 min) and 259 min (100–685 min), respectively. In our hospital, MISt surgery was associated with a mean operative time of 159.8 ± 65.4 min (52–410 min), a mean blood loss of 115.7 ± 126.7 g (10–630 g), and a mean duration of postoperative bed rest of 3.2 ± 2.2 days (1–14 days). It had also been reported that MISt has lower transfusion volume and fewer complications compared to the conventional spinal surgery [18,19]. This indicates that MISt can be used in more cases, even in elderly patients and those with a prognosis of ≤6 months. In other words, MISt will open up opportunities for patients who are ineligible for conventional surgical methods because of their high invasiveness.MISt results in early postoperative ambulation and early discharge. This is a very helpful benefit for patients with metastatic spinal tumors because their life expectancy is limited. Miscusi [16] reported that the average length of hospital stay was 7.2 days in the MISS group (*p* < 0.01) and 9.25 days in the conventional method group. Cui also reported that the average length of hospital stay for conventional and MIS surgery was 8.8 and 11.3 days, respectively [19]. At our hospital, 31 of 51 patients were discharged home after surgery (mean 31.9 ± 18.0 days after surgery), and 17 were transferred to another institution (mean 39.4 ± 7.0 days after surgery). The reason why our hospital stay was longer than other reports was that the primary physician for the hospital stay was basically the oncologist, not the orthopedic surgeon, due to multidisciplinary treatment [6]. The advantage was that the oncology department could start treatment as soon as possible, while the primary physician had a long hospital stay to introduce other adjuvant therapies after the surgery. Furthermore, because increasing numbers of adjuvant therapies are now being administered or performed on an outpatient basis, we believe that MISt provides a great advantage by allowing early discharge from the hospital.MISt allows fixation with less contamination and does not interfere with other adjuvant therapies. If only fixation is performed, contamination is almost irrelevant because the affected area is not expanded. In addition, there is less damage to soft tissues and so forth, and early transition to anticancer drugs and radiation therapy is possible. If decompression is not added, the patient can be immobilized so as to avoid the irradiated area, thus preventing complications such as skin problems, even if radiotherapy is started early after surgery. Zairi et al. [20] wrote that the most important benefit of using minimally invasive approaches is the more rapid initiation of postoperative adjuvant therapies. Conventional methods require wide exposure, resulting in a large dead space and tissue necrosis. For these reasons, Zairi et al. [20] stated that adjuvant therapy should not be started until 4–5 weeks after surgery to prevent wound dehiscence and infection. At our hospital, we consider that MISt can avoid complications such as skin problems even if radiotherapy is started early after surgery, because it can be fixed in a way that avoids areas that can be irradiated. Moreover, if detected early through multidisciplinary treatment, interventions can be made before nerve compression or pathological fractures occur, thus eliminating the need for decompression surgery.

On the other hand, there are some disadvantages of MISt for metastatic spinal tumor:MISt carries risks and is dependent on surgeon skill. It is challenging to become skilled in percutaneous surgery; however, this expertise is necessary to achieve successful and safe outcomes. Furthermore, radiation exposure poses a risk to both the patients and surgeon.Bone transplantation is impossible with MISt. Atanasiu et al. [21] reported that bone transplantation is required for patients with a life expectancy of 2 years or longer. In cases of long-term survival contrary to the prognosis, it is necessary to consider whether additional bone grafting should be performed. However, in the present study, only 2 of the 51 patients underwent reoperation due to loosening of the screws. The results of adjuvant therapy have been improved by advances in radiotherapy, bisphosphonates, and various chemotherapies. Bellato [22] reported that 9 of 105 (8.57%) patients with cancer who underwent spinal fusion experienced mechanical failure; the mean time to failure was 9.5 months, and the most common failure was implant loosening. Furthermore, the mean and median durations of survival were 22.76 and 7.4 months, respectively. Thus, 88% of patients had a shorter survival duration than the average time required to develop mechanical complications. The authors noted that no cases required further surgery. In our opinion, temporary fixation with MISt is appropriate until unstable lesional vertebral bodies are remodeled and become stable.Even MISt is associated with postoperative pain. In many cases, patients with cancer experience severe preoperative pain and are receiving high doses of morphine. MISt is a minimally invasive procedure; however, controlling postoperative pain is often difficult.

As discussed above, MISt is a useful treatment for metastatic spinal tumors. It provides hope to patients who were previously ineligible for treatment because they did not qualify for conventional invasive surgeries. Uei et al. [13] reported the results of multidisciplinary treatment using palliative posterior spinal stabilization surgery, and the median postoperative survival time was 12 months. We used the same combination of multidisciplinary treatment and spinal stabilization as Uei et al. [13]. The median survival time in our study was 22.0 ± 8.6 months in our study. MISt also allows for early hospital discharge, which is beneficial given the current shift of cancer treatment to outpatient settings.

On the other hand, the indications for surgery are also important. We should avoid choosing surgery easily because it is less invasive. Although there are some reports that an interdisciplinary discussion was carried out and decide on surgery before surgery [18], most of the indications for surgery are intolerable back pain, neurologic deficits, compression, spinal instability, and resistance to conservative treatment [13,23]. There are many reports using SINS to evaluate spinal instability, but the indications for SINS in cases of imminent instability of 7 to 12 points are ambiguous and depend largely on the surgeon’s preference. In our hospital, not only SINS, but also further strict indications, were set and surgery was performed, but it is necessary to study the indications of MISt for this disease with more cases.

Furthermore, surgery for appropriate patients at appropriate times prolongs life expectancy, even in the context of spinal metastases. Watanabe et al. [17] and Helweg-Larsen et al. [24] reported that the severity of preoperative paralysis had a significant impact on final walking ability. After paralysis has occurred, the results are not good, even if the operation is performed with MISt. It is also important to detect the disease at an early stage by using multidisciplinary treatment, as in this study. Our results are due to the fact that we intervene early in all metastatic spine tumors [6]. Rather than providing palliative or urgent treatment to all patients, it is advisable to detect candidate patients as early as possible and to restore spine-related functions with less invasive surgery before these functions are lost and neurological symptoms occur.

### Limitations

Future issues to be addressed include the small number of cases in the surgery group and the lack of analysis of each type of carcinoma. The lack of a control in this study is also considered a limitation.

## 5. Conclusions

MISt for metastatic spinal tumors is a useful surgery, particularly in light of the recent shift to outpatient cancer treatment;MISt can improve the outcome of patients who are ineligible for conventional, highly invasive surgery;Choosing the right surgery for the right patient at the right time has a significant impact on life expectancy.

## Figures and Tables

**Figure 1 medicina-58-00358-f001:**
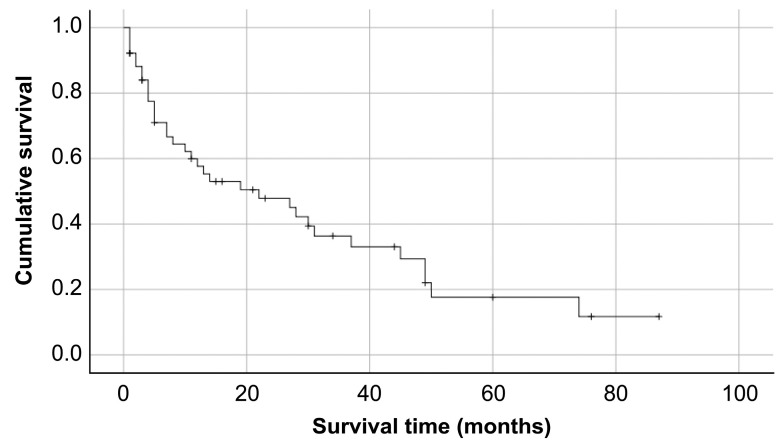
Kaplan–Meier curve for survival in MISt, 22.0 ± 8.6 months (95% confidence interval [CI] 5.1–38.8).

**Figure 2 medicina-58-00358-f002:**
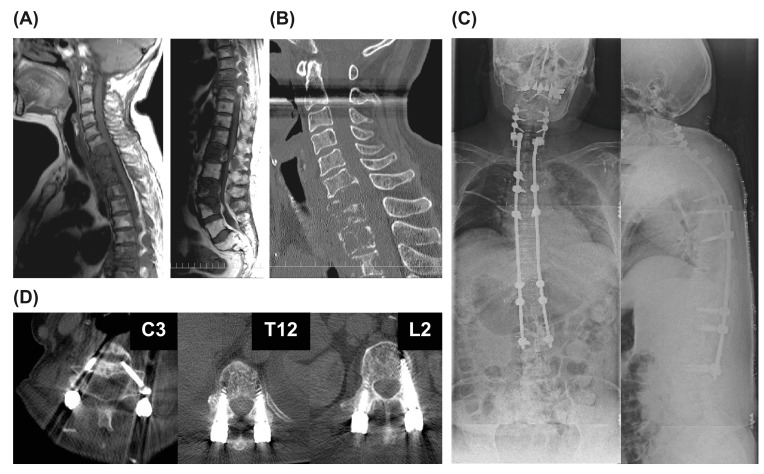
Case. A 63-year-old woman with metastatic spinal tumor of breast cancer. (**A**) Radiographs showing the first augmentation surgery. (**B**) Preoperative computed tomography (CT) image. (**C**) Radiograph showing the whole spine after surgery. (**D**) CT image at 43 months after surgery (C3, T12, L2 level).

**Table 1 medicina-58-00358-t001:** Detailed clinical characteristics and results of patients with metastatic spine tumors who underwent MISt surgery.

	MISt
Patients, n	51
Gender	Male 34; female 17
Age at diagnosis of metastatic spinal tumors, years	68.9 ± 9.9
Postoperative follow-up period, months	20.8 ± 21.8
Metastatic tumor diagnosis, n (%)	
Breast cancer	11 (21.6)
Lung cancer	10 (11.4)
Prostate cancer	7 (4.7)
Gastrointestinal cancer	5 (9.8)
Renal pelvis/ureter cancer	4 (7.8)
Kidney cancer	3 (5.9)
Thyroid cancer	2 (3.9)
Uterine cancer	2 (3.9)
Multiple myeloma	2 (3.9)
Primary unknown cancer	2 (3.9)
Pancreatic cancer	1 (2.0)
Head and neck cancer	1 (2.0)
Others	1 (2.0)
SINS	9.4 ± 2.8
Tokuhashi score	9.3 ± 2.6
New Katagiri score	4.6 ± 2.0
PS (preop)	2.5 ± 1.3
Number of deaths, n (%)	35 (68.6%)
Adjuvant therapy	
Radiotherapy (RT)	46 (90.2%)
Bone-modifying agents	41 (80.4%)

SINS, spinal instability neoplastic score^7)^; PS, performance status. PS was evaluated using the Eastern Cooperative Oncology Group Performance status (ECOG PS) scale^12)^.

**Table 2 medicina-58-00358-t002:** Details of the surgery.

Operative time (minutes)	159.8 ± 65.4
Blood loss (mL)	115.7 ± 126.6
Fixed segments (vertebrae)	6.2
Additional decompression	14
Ambulation (days)	3.2 ± 2.2
PS (preop)	2.5 ± 1.3
PS (postop)	1.7 ± 1.3
Discharge (cases)	31
Transferred to another hospital (cases)	17

PS, performance status. PS was evaluated using the Eastern Cooperative Oncology Group Performance status (ECOG PS) scale^12)^.

**Table 3 medicina-58-00358-t003:** Details of preoperative SINS evaluation and additional laminectomy.

SINS	Assessment of Spinal Instability	Cases	Additional Decompression, n (%)
0–6	Stability	7	2 (28.6)
7–12	Imminent instability	38	8 (21.1)
≥13	Instability	6	4 (66.7)

SINS, spinal instability neoplastic score^7)^.

**Table 4 medicina-58-00358-t004:** Improvement of paralysis by Frankel’s classification before and after surgery.

Frankel	A	B	C	D	E
A	2			1	
B		0			
C			3	1	
D				5	2
E					37

The vertical axis is the preoperative Frankel’s classification, and the horizontal axis is the postoperative Frankel’s classification.

**Table 5 medicina-58-00358-t005:** Cases of early death.

Cases	1	2	3	4	5	6	7	8
Age	0	72	46	81	62	63	79	67
Gender	M	M	F	M	M	M	M	M
Primary Tumors	Esophageal cancer	Renal pelvis cancer	Uterine sarcoma	Multiple myeloma	Primary unknown cancer	Colon cancer	Lung cancer	Gastric cancer
SINS	14	9	3	7	12	16	10	10
Frankel grade	D	E	E	E	D	E	E	E
Tokuhashi	7	13	13	8	6	11	5	7
New Katagiri	9	3	2	2	6	7	8	5
PS (pre-op)	3	0	2	3	4	3	3	1
PS (post-op)	4	0	0	4	3	3	1	1
Postoperative period	1	1	1	1	2	2	3	3
Consult to orthopedic surgery	Detected by imaging tests	Detected by imaging tests	Detected by imaging tests	After the onset of SRE	After the onset of SRE	After the onset of SRE	After the onset of SRE	Detected by imaging tests
Outcome	*	*	25 days transferred to another hospital	*	35 days transferred to another hospital	39 days discharged home	31 days discharged home	20 days discharged home
Cause of death	Brain metastasis	Sepsis	Deterioration of general condition	Aspiration pneumonia	Deterioration of general condition	Liver metastasis	AMI	Deterioration of general condition

* The patient died during hospitalization.

**Table 6 medicina-58-00358-t006:** Mortality by cancer type.

Primary Tumors	Number of Cases	Number of Deaths	Mortality Rate (%)	Postoperative Period (Months)
Breast cancer	11	5	45.4	39.6
Lung cancer	10	9	90	16.5
Prostate cancer	7	2	28.6	27.7
Gastrointestinal cancer	5	5	100	2.4
Ureter cancer	4	3	75	9.3
Kidney cancer	3	1	33.3	38.7
Thyroid cancer	2	2	100	24.5
Uterine cancer	2	2	100	7
Multiple myeloma	2	2	100	6.5
Primary unknown cancer	2	1	50	1.5
Pancreatic cancer	1	1	100	5
Head and neck cancer	1	1	100	1
Others	1	1	100	4

## Data Availability

Not applicable.

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
