# Peer review of "Effect of Minimally Invasive Spine Stabilization in Metastatic Spinal Tumors"

_medicina, 2022, doi:10.3390/medicina58030358_

Round 1
Reviewer 1 Report
This is an interesting paper about Minimally Invasive Spine Stabilization (MISt) in Metastatic Spinal Tumors. The paper is well designed and written.
However, I think that there are several concerns that should be made before publication.
1 Line 37
It would be better to list them as bone-modifying agents because there were anti-RANKL antibodies as well as bisphosphonates alone.
In the context of molecular-targeted drugs, immune checkpoint inhibitors.
2 Line 63
Was MISt performed in all cases of bone metastases during this period, or no TES?
3Line 115 Table list
â‘ Wouldn't it be easier to visualize if the tumors were aligned on the right-hand side rather than the center?
â‘¡The n (%) should be written next to the Number of deaths
â‘¢Follow-up period should also be listed in units (months?).
â‘£No cases of use of anti-RANKL antibodies?
ï¼”Lineï¼’ï¼—ï¼’
>"only 2 of the 51 patients underwent reoperation due to loosening of the screw"
This content is not in Methods or Results?
5 Line283
> Even MISt is associated with postoperative pain. In many cases, patients with cancer experience severe preoperative pain and are receiving high doses of morphine. MISt is a minimally invasive procedure; however, controlling postoperative pain is often difficult.
I completely agree with you, but isn't it less painful than the conventional method?
5 Line286-302
I think line 268 onwards corresponds to advantages of MISt, not disadvantages of MISt.
6 Line 304
The lack of control in retrospective studies, as described in the STROBE checklist, is also considered to be a limitation.
Author Response
February 11, 2022
Dr. Lily Fan
Special Issue Editor
Medicina
Dear Dr. Lily Fan:
Thank you for providing us the opportunity to submit a revised draft of the manuscript “Effect of Minimally Invasive Spine Stabilization in Metastatic Spinal Tumors” for publication in Medicina. We appreciate the time and effort that you and the reviewers have dedicated in providing feedback on our manuscript and are grateful for the insightful comments valuable improvements suggested to further improve the quality of our manuscript. We have incorporated most of the suggestions made by the reviewers, and those changes have been denoted using tracks within the manuscript. Please see below, in blue, our point-by-point responses to the reviewers’ comments and concerns. All page numbers refer to the revised manuscript file with tracked changes.
Reviewers’ Comments to the Authors:
Reviewer 1
This is an interesting paper about Minimally Invasive Spine Stabilization (MISt) in Metastatic Spinal Tumors. The paper is well designed and written.
Author response: Thank you for your valuable and positive feedback.
However, I think that there are several concerns that should be made before publication.
1 Line 37
It would be better to list them as bone-modifying agents because there were anti-RANKL antibodies as well as bisphosphonates alone.
Author response: Thank you for pointing this out. We have revised it accordingly.
In the context of molecular-targeted drugs, immune checkpoint inhibitors.
Author response: Thank you for this comment. Molecular-targeted drugs and immune checkpoint inhibitors are shown separately in the treatment of lung cancer. Therefore, we have left them as they are.
2 Line 63
Was MISt performed in all cases of bone metastases during this period, or no TES?
Author response: All surgeries performed during this period were MISt only, and TES was not performed in any case. We checked all metastatic spine tumors as soon as they were detected, and only few indications were observed for TES.
3Line 115 Table list
â‘ Wouldn't it be easier to visualize if the tumors were aligned on the right-hand side rather than the center?
Author response: Thank you for pointing this out. We have made the corrections as suggested.
â‘¡The n (%) should be written next to the Number of deaths
Author response: Thank you for pointing this out. We have revised it accordingly.
â‘¢Follow-up period should also be listed in units (months?).
Author response: Thank you for pointing this out. We have revised it accordingly.
â‘£No cases of use of anti-RANKL antibodies?
Author response: No anti-RANKL antibodies were used in this case. The treatment method was changed to bone-modifying agents. In addition, “RT” has been changed to “radiotherapy (RT),” and lines 145-6 have also been changed.
ï¼”Lineï¼’ï¼—ï¼’
>"only 2 of the 51 patients underwent reoperation due to loosening of the screw"
This content is not in Methods or Results?
Author response: This content has been written in lines 140–142.
5 Line283
> Even MISt is associated with postoperative pain. In many cases, patients with cancer experience severe preoperative pain and are receiving high doses of morphine. MISt is a minimally invasive procedure; however, controlling postoperative pain is often difficult.
I completely agree with you, but isn't it less painful than the conventional method?
Author response: Thank you for this suggestion. It would have been interesting to explore this aspect. Of course, MISt is less painful than the conventional method, but it does not respond well to painkillers and there are considerable complaints of pain. Moreover, the pain is more severe than after MISt surgery for another disease.
5 Line286-302
I think line 268 onwards corresponds to advantages of MISt, not disadvantages of MISt.
Author response: Although we think this is an excellent suggestion, we respectfully disagree with it. At present, there are no devices available for MISt implants that allow for bone grafting, and many studies have indicated that implants can loosen. However, in our opinion, temporary fixation with MISt is appropriate until unstable lesional vertebral bodies are remodeled and become stable (lines 280–282).
6 Line 304
The lack of control in retrospective studies, as described in the STROBE checklist, is also considered to be a limitation.
Author response: Thank you for this excellent suggestion. We agree that this is a potential limitation of the study and have added it to the Limitation section.

Reviewer 2 Report
Thank you for giving me the opportunity to review this maunscript
Major:
- The manuscript provides limited new data. However, similar results in more extensive or comparable cohorts have been published previously (e.g., Zhu et al. 2021 Spine; or Cui et al. 2021 Cancer Management and Research; studies that should be discussed).
- The frequency of MISt operations seems low, with a calculated average interval between two operations of approximately 50 days. However, the prevalence of spinal metastasis is a relatively common condition. How can the authors explain this discrepancy?
- Table 5 is hard to read and does not give important additional information. Please better summarize the essential points of this table
- How do you explain the longer hospital stay of your patients compared to other minimal invasive fixation studies (e.g., see the studies as mentioned earlier)?
- Please reduce the number of used prognostic scores. For example, I cannot see the point of using the Tokuhashi score and the Tomita score or the Katagiri score and the New Katagiri score.
Minor:
- Please use the numeric notation of numbers greater than 12 (e.g., “fifty-one” see Abstract).
- Why is blood loss documented in Gramm?
- Please keep it consistent. Either use abbreviations according to the Si base unit system (see “g” in the abstract) if applicable or always write out the units (see minutes or days in the abstract).
- Please list the tumor entities strictly in a decreasing order except for “others” in the Result section.
- Why are tumor entities listed in Table 1 that were not detected? I would suggest removing them to save space.
- Table 3 last row I guess it should be “≥13” instead of “13”.
Author Response
February 11, 2022
Dr. Lily Fan
Special Issue Editor
Medicina
Dear Dr. Lily Fan:
Thank you for providing us the opportunity to submit a revised draft of the manuscript “Effect of Minimally Invasive Spine Stabilization in Metastatic Spinal Tumors” for publication in Medicina. We appreciate the time and effort that you and the reviewers have dedicated in providing feedback on our manuscript and are grateful for the insightful comments valuable improvements suggested to further improve the quality of our manuscript. We have incorporated most of the suggestions made by the reviewers, and those changes have been denoted using tracks within the manuscript. Please see below, in blue, our point-by-point responses to the reviewers’ comments and concerns. All page numbers refer to the revised manuscript file with tracked changes.
Reviewers’ Comments to the Authors:
Reviewer 2
Thank you for giving me the opportunity to review this manuscript
Major:
- The manuscript provides limited new data. However, similar results in more extensive or comparable cohorts have been published previously (e.g., Zhu et al. 2021 Spine; or Cui et al. 2021 Cancer Management and Research; studies that should be discussed).
Author response: Thank you for this suggestion. In this study, we mainly focused on the effects of MISt for metastatic spinal tumor and also tried to explore the surgical indications of MISt . Although performing MISt surgery in patients was useful, early detection and treatment before the onset of SREs as well as the combination of MISt surgery with other treatments also proved beneficial. If MISt surgery is performed only due to its minimally invasive nature, only the medical costs will increase. In future, studies on more number of cases should be performed to evaluate the surgical indications of MISt.
- The frequency of MISt operations seems low, with a calculated average interval between two operations of approximately 50 days. However, the prevalence of spinal metastasis is a relatively common condition. How can the authors explain this discrepancy?
Author response: Thank you for this suggestion. While we appreciate this feedback, we respectfully disagree with it.
MISt surgery is useful for metastatic spinal tumors. However, the opinion that MISt should be performed because it is a minimally invasive procedure is problematic.
Indications for surgery are important.
The problem with metastatic spinal tumors is that unlike other bone metastases, when SREs occur, they not only reduce QOL but also affect other treatments. If QOL decreases, anticancer drug therapy and radiation therapy are no longer indicated. Ultimately, it can affect life expectancy. Intervention by an orthopedic surgeon as soon as bone metastasis is detected, such as the multidisciplinary treatment we provide, can prevent the occurrence of SREs. Therefore, if SREs are detected before they occur, there is a possibility that surgery itself may not be necessary due to advances in other treatments.
We think that our study makes a valuable contribution to the field.
Table 5 is hard to read and does not give important additional information. Please better summarize the essential points of this table
Author response: Thank you for this suggestion. Table 5 has been corrected.
- How do you explain the longer hospital stay of your patients compared to other minimal invasive fixation studies (e.g., see the studies as mentioned earlier)?
Author response: Thank you for this suggestion. In our multidisciplinary treatment, the primary care physician was experienced in treating carcinomas. Orthopedic surgeons provided support for management and surgery. Because cancer is a systemic disease, surgery is a part of treatment, and systemic treatment should be supervised by the main department treating the cancer. As a result, the burden on orthopedic surgeons can be reduced. After the surgery, chemotherapy, radiation therapy, and cancer treatment planning are performed by the attending physician, increasing the length of hospital stay.
- Please reduce the number of used prognostic scores. For example, I cannot see the point of using the Tokuhashi score and the Tomita score or the Katagiri score and the New Katagiri score.
Author response: The Tomita score and Katagiri score were excluded.
Minor:
- Please use the numeric notation of numbers greater than 12 (e.g., “fifty-one” see Abstract).
Author response: We have revised it accordingly (Lines14, 23).
- Why is blood loss documented in Gramm?
Author response: We have replaced “gm” with “ml” at all instances in the manuscript.
- Please keep it consistent. Either use abbreviations according to the Si base unit system (see “g” in the abstract) if applicable or always write out the units (see minutes or days in the abstract).
Author response: We have revised it as per your suggestion.
- Please list the tumor entities strictly in a decreasing order except for “others” in the Result section.
Author response: We have revised it as per your suggestion (Lines107–111).
- Why are tumor entities listed in Table 1 that were not detected? I would suggest removing them to save space.
Author response: Thank you for this suggestion. We have deleted the entities that were not detected.
- Table 3 last row I guess it should be “≥13” instead of “13”.
Author response: Thank you for pointing this out. We have corrected it accordingly.

Reviewer 3 Report
This is a well written overview of the authors minimally invasive treatment strategy in spinal metastatic disease.Author Response
February 11, 2022
Dr. Lily Fan
Special Issue Editor
Medicina
Dear Dr. Lily Fan:
Thank you for providing us the opportunity to submit a revised draft of the manuscript “Effect of Minimally Invasive Spine Stabilization in Metastatic Spinal Tumors” for publication in Medicina. We appreciate the time and effort that you and the reviewers have dedicated in providing feedback on our manuscript and are grateful for the insightful comments valuable improvements suggested to further improve the quality of our manuscript. We have incorporated most of the suggestions made by the reviewers, and those changes have been denoted using tracks within the manuscript. Please see below, in blue, our point-by-point responses to the reviewers’ comments and concerns. All page numbers refer to the revised manuscript file with tracked changes.
Reviewers’ Comments to the Authors:
Reviewer 3
This is a well written overview of the authors minimally invasive treatment strategy in spinal metastatic disease.
Author response: Thank you for your comments.

Round 2
Reviewer 2 Report
I approve an improvement of the manuscript done by the authors. But, unfortunately, some critics were not sufficiently revised.
- Regarding the former comment 2: I think there is a misunderstanding. My question is: Why did the authors only operate an average of 7 patients per year using MISt?
- The authors were kindly asked to better embed their study into the current literature regarding minimally invasive stabilization for metastatic spinal disease (e.g., DOI: 10.1097/BRS.0000000000001893; 10.2147/CMAR.S332985, 0.1016/j.wneu.2020.12.130) and explain the novelties cp. to these studies.
- On a second look through the references, an inappropriate amount of self-citation is suspected without a direct connection to the presented study.
Additional comment:
I cannot review the new version of Table 5 due to the amounts of changes being tracked.
Author Response
Dear Dr. Lily Fan:
Thank you for providing us the opportunity to submit a revised draft of the manuscript “Effect of Minimally Invasive Spine Stabilization in Metastatic Spinal Tumors” for publication in Medicina. We appreciate the time and effort that you and the reviewers have dedicated in providing feedback on our manuscript and are grateful for the insightful comments valuable improvements suggested to further improve the quality of our manuscript. We have incorporated most of the suggestions made by the reviewers, and those changes have been denoted in red font within the manuscript. Please see below, in blue, our point-by-point responses to the reviewers’ comments and concerns. All page numbers refer to the revised manuscript file with tracked changes.
Reviewers’ Comments to the Authors:
Reviewers’ Comments to the Authors:
Reviewer 2
Thank you for giving me the opportunity to review this maunscript
Major:
- Regarding the former comment 2: I think there is a misunderstanding. My question is: Why did the authors only operate an average of 7 patients per year using MISt?
The authors were kindly asked to better embed their study into the current literature regarding minimally invasive stabilization for metastatic spinal disease (e.g., DOI: 10.1097/BRS.0000000000001893; 10.2147/CMAR.S332985, 0.1016/j.wneu.2020.12.130) and explain the novelties cp. to these studies.
On a second look through the references, an inappropriate amount of self-citation is suspected without a direct connection to the presented study.
Author response: Thank you for your valuable and positive feedback.
We have picked up all the patients diagnosed with metastatic spine tumor in our hospital in a new multidisciplinary treatment as described in the literature12,13. During the period of the current study, December 2013 to October 2020, we found 992 patients with spinal metastases. The patients were discussed at conferences, and a total of 51 patients required surgery, about 7 per year. The indications for surgery are described in lines 66-78, and surgery is indicated in patients with SINS 13 points or more, tumors at the level of the thoracic spine in 7-12 points, vertebral bodies already crushed, and vertebral arch roots and posterior walls dissolved. This is the indication for surgery. In the 992 cases, there were no patients with general indications for TES. This means that patients with spinal metastases do not need surgery if detected early, and radiotherapy and chemotherapy are sufficient.
There was also no patient who underwent emergency surgery for SRE (paralysis or sickle fracture) during this period. This study demonstrates the benefits of surgery for patients with metastatic spine tumors and also focuses on the indications for surgery (the role of surgery) in multidisciplinary treatment. We believe that this study warns against the recent trend in this field of minimally invasive surgery. Therefore, we believe that our references 12.13 are not unnecessary, but necessary to show the multidisciplinary treatment.
・I cannot review the new version of Table 5 due to the amounts of changes being tracked.
Author response: Thank you. Table 5 shows the original form, with the prognostic score narrowed down to two.
|
Table5 |
||||||||
|
Cases |
1 |
2 |
3 |
4 |
5 |
6 |
7 |
8 |
|
Age |
0 |
72 |
46 |
81 |
62 |
63 |
79 |
67 |
|
Gender |
M |
M |
F |
M |
M |
M |
M |
M |
|
Primary Tumors |
Esophageal |
Renal pelvis |
Uterine |
Multiple |
Primary |
Colon |
Lung |
Gastric |
|
SINS |
14 |
9 |
3 |
7 |
12 |
16 |
10 |
10 |
|
Frankel grade |
D |
E |
E |
E |
D |
E |
E |
E |
|
Tokuhashi |
7 |
13 |
13 |
8 |
6 |
11 |
5 |
7 |
|
New Katagiri |
9 |
3 |
2 |
2 |
6 |
7 |
8 |
5 |
|
PS (pre-op) |
3 |
0 |
2 |
3 |
4 |
3 |
3 |
1 |
|
PS (post-op) |
4 |
0 |
0 |
4 |
3 |
3 |
1 |
1 |
|
Postoperative period |
1 |
1 |
1 |
1 |
2 |
2 |
3 |
3 |
|
Consult to |
Detected by |
Detected by |
Detected by |
After the |
After the |
After the |
After the |
Detected by |
|
Outcome |
* |
* |
25days transferred |
* |
35days transfered |
39days |
31days |
20days |
|
Cause of death |
Brain |
Sepsis |
Deterioration of |
Aspiration |
Deterioration of |
Liver |
AMI |
Deterioration of |

Round 3
Reviewer 2 Report
Two comments are still not commented:
The authors were kindly asked to better embed their study into the current literature regarding minimally invasive stabilization for metastatic spinal disease (e.g., DOI: 10.1097/BRS.0000000000001893; 10.2147/CMAR.S332985, 0.1016/j.wneu.2020.12.130) and explain the novelties cp. to these studies.
On a second look through the references, an inappropriate amount of self-citation is suspected without a direct connection to the presented study.
Author Response
Reviewers’ Comments to the Authors:
Reviewer 2
・The authors were kindly asked to better embed their study into the current literature regarding minimally invasive stabilization for metastatic spinal disease (e.g., DOI: 10.1097/BRS.0000000000001893; 10.2147/CMAR.S332985, 0.1016/j.wneu.2020.12.130) and explain the novelties cp. to these studies.
Author response: Thank you for your valuable and positive feedback.
We have incorporated the new literature that you pointed out and described the advantages of this study.
We have added the advantages of MISt (lines 249-260), the importance of surgical indications (lines 70-73, 309-318), the warning bell in easy MISt surgery (lines 309-310), and the importance of combining MISt with multidisciplinary treatment (lines 274-276, 303-305, 323-324).
On a second look through the references, an inappropriate amount of self-citation is suspected without a direct connection to the presented study.
Author response: Thank you for pointing this out.
The references have been carefully examined and organized again.